# The Impact of Hyaluronic Acid on Tendon Physiology and Its Clinical Application in Tendinopathies

**DOI:** 10.3390/cells10113081

**Published:** 2021-11-09

**Authors:** Francesco Oliva, Emanuela Marsilio, Giovanni Asparago, Antonio Frizziero, Anna Concetta Berardi, Nicola Maffulli

**Affiliations:** 1Department of Musculoskeletal Disorders, Faculty of Medicine and Surgery, University of Salerno, 84084 Baronissi, Italy; emanuelamarsilio16@gmail.com (E.M.); giovanni.asparago@gmail.com (G.A.); n.maffulli@qmul.ac.uk (N.M.); 2Clinica Ortopedica, Ospedale San Giovanni di Dio e Ruggi d’Aragona, 84131 Salerno, Italy; 3Physical and Rehabilitation Medicine Unit, Department of Medicine and Surgery, University of Parma, 43126 Parma, Italy; antonio.frizziero@unipr.it; 4Department of Haematology, Laboratory of Stem Cells, Transfusion Medicine and Biotechnologies, Santo Spirito Hospital, 65123 Pescara, Italy; annacberardi@yahoo.it; 5Centre for Sports and Exercise Medicine, Barts and the London School of Medicine and Dentistry, Queen Mary University of London, Mile End Hospital, 275 Bancroft Road, London E1 4DG, UK; 6School of Pharmacy and Bioengineering, Keele University Faculty of Medicine, Thornburrow Drive, Stoke on Trent ST4 7QB, UK

**Keywords:** hyaluronic acid, receptor, biology, structure, tendon, tendinopathy, degeneration, inflammation, viscoelastic, hygroscopic, effect

## Abstract

The physical–chemical, structural, hydrodynamic, and biological properties of hyaluronic acid within tendons are still poorly investigated. Medical history and clinical applications of hyaluronic acid for tendinopathies are still debated. In general, the properties of hyaluronic acid depend on several factors including molecular weight. Several preclinical and clinical experiences show a good efficacy and safety profile of hyaluronic acid, despite the absence of consensus in the literature regarding the classification according to molecular weight. In in vitro and preclinical studies, hyaluronic acid has shown physical–chemical properties, such as biocompatibility, mucoadhesivity, hygroscopicity, and viscoelasticity, useful to contribute to tendon healing. Additionally, in clinical studies, hyaluronic acid has been used with promising results in different tendinopathies. In this narrative review, findings encourage the clinical application of HA in tendinopathies such as rotator cuff, epicondylitis, Achilles, and patellar tendinopathy.

## 1. Introduction

Hyaluronic acid (HA) is a non-sulphated glycosaminoglycan formed by repetitive units of glucuronic acid and N-acetyl glucosamine. HA is widely present in the extracellular matrix in vertebrates and invertebrates to confer mechanical support, viscoelastic and hygroscopic properties, and anti-inflammatory effects to cells and tissue. HA is one of the fundamental components of cartilage and tendon tissue, contributing to their viscoelastic properties [1,2,3,4,5]. HA enhances the cellular activities of fibroblasts, including their adhesivity, extracellular matrix (ECM) synthesis, and proliferation, but, despite its influence on cells and tendon structure, its role on the biomechanical function is still not clarified [6,7]. The ECM of tendons is predominantly composed of type I collagen (Figure 1) and proteoglycans, with a small amount of the other types of collagen (type II, III, V, VI, IX, XI) (Table 1). The predominant cells type within the tendon are tenoblasts or tenocytes, accounting for 90–95% of cells present in tendon tissue between collagen fibres [8]. Tenoblasts and tenocytes show different metabolic features: while tenoblasts have high activity, tenocytes are metabolically less active. Both types of cells produce collagen, elastin, ECM, and proteins [9]. Other types of cells present in lesser quantities include chondrocytes at the bone attachment and insertion sites, synovial cells in the tendon sheath, and vascular cells, capillary endothelial cells, and arterioles’ smooth muscle cells. All cells that produce the ECM are involved in endogenous HA production as well. HA injections are widely used to treat osteoarthritis (OA), but their efficacy in the management of tendinopathies is still debated [10]. Tendinopathies are characterised by tendon structure disruption, ineffective neovascularisation, decreased collagen I, and enhanced collagen II production [10]. Recently, the anti-inflammatory and viscoelastic effects of HA on connective tissue have been suggested to warrant the use of HA for the treatment of tendinopathies [11]. Some studies proved and support its use to improve function and reduce pain in tendinopathies [12,13], avoiding the complications of corticosteroids [14]. This review outlines the role of HA in reducing inflammation and increasing regeneration in tendinopathies in order to better understand its clinical applications.

## 2. HA Synthesis, Properties, and Degradation

HA is synthesised by a class of three integral membrane proteins named HA synthases (HAS1, HAS2 and HAS3). They synthetise HA by repeated addition of glucuronic acid and N-acetyl-D-glucosamine groups, and HA molecules are exported through transporters through the cell wall into the cells [15]. HAS1 and HAS2 proteins are moderately active and responsible for the synthesis of high molecular weight HA (HMW-HA) (>500 kDa), whereas HAS3, with high activity, synthetises low molecular weight HA (LMW-HA) ranging from 20 to 450 kDa (Table 2). 

HA synthesis is reversed by a 4-methylumbelliferone (hymecromone, heparvit), a 7-hydroxy-4-methylcoumarin derivate [16]. The degradation of tendon collagen fibrils is mainly regulated by a complex system of metalloproteinases (MMP), which self-regulates by means of specific inhibitors (TIMP). The activity of this system is in relation to the levels of the mechanical structure of the tendon. Historically, in some in vitro experimental animal models, HA does not directly block the production of MMPs or increase the synthesis of its inhibitors by itself [17]. However, HA is likely able to regulate the activity of some MMPs through a downregulation mechanism [18]. In experimental studies in humans, HA is also able to block the action of fibronectin fragments (a molecule synthesised during tendon repair capable of activating MMP), inhibiting degradative processes and promoting repair and biosynthesis of the molecular constituents of the tendon fascicles [19]. HA acts on the ECM through specific and non-specific interactions with several receptors (Table 3). 

Some examples of receptors and molecules that show an interaction with HA are CD44, the receptor for hyaluronan-mediated motility (RHAMM), lymphatic vessel endothelial hyaluronan receptor 1 (LYVE-1), Toll-like receptors 2-4 (TLR2-4), TNF-stimulated gene 6 (TSG6), glial hyaluronate-binding protein (GHAP), versican, aggrecan, and neurocan [20]. Versican, aggrecan, and neurocan are HA-bound chondroitin sulphate and keratan sulphate proteoglycans that can be found in the ECM of different tissues. Neurocan is mainly located in the nervous tissue; it is involved in cell adhesion and plays an important role in local axonal growth [21]. Aggrecan and versican also regulate cell growth and are involved in cell migration and haemostasis [22,23]. Aggrecan is mainly located in the cartilage, and it plays a key role in joint cartilage function and in the development of OA [23]. Versican is involved in skeletal development, cardiovascular, and nervous morphogenesis during embryogenesis [24]. Aggrecan and versican, linked to HA, allow the tendon to acquire a high resistance to compression and traction forces due to loading and mobilisation [25]. RHAMM is a protein located in the cytoplasm, nucleus, and plasma membrane [26]. The nuclear receptor for hyaluronan mediated motility (Nuclear-RHAMM) regulates cell motility and inflammation [27]. In the cytoplasm, RHAMM interacts directly or indirectly via binding proteins with actin filaments and microtubules, regulating cell polarity and cell migration [27,28,29]. Extracellular RHAMM acts on cell transformation and migration during the HA-dependent tissue damage and repair process [2]. Extracellularly, it acts with CD44 [27]. RHAMM is expressed in several healthy and neoplastic human tissues. Greater expression of the RHAMM protein in tumour tissues is associated with a higher degree of malignancy [30,31,32,33,34]. In healthy adult tissues, the RHAMM protein shows limited expression. RHAMM has been found in the thymus, lymph nodes, small intestine, colon, skin, and bone marrow [35]. LYVE-1 is a receptor found on endothelial cells of lymphatic tissue that binds to HA. This protein is involved in the lymphatic transport of leukocytes [36]. Toll-like receptors 2 and 4 (TLR-2 and TLR-4) are receptors expressed on chondrocytes and peritendinous cells [37,38]. It has been observed that the expression of TLR-4 is increased on the cell surface of chondrocytes in patients with rheumatoid arthritis, osteoarthritis, and tendinopathies [38,39]. LMW-HA binds to TLR-4 producing an inflammatory response in the joint, while HMW-HA reduces this stimulation hiding the active site of TLR-4 [40]. TSG-6 is a secreted protein with immunomodulating action for mesenchymal/stromal stem cells. The connection among HA and TSG-6 leads to anti-inflammatory processes and protective action on the tissues [41]. Glial hyaluronate-binding protein (GHAP) is an ECM protein of nervous tissues that reduces the spread of the inflammatory process [42]. CD44 regulation plays a fundamental role in lymphocyte cell stimulation and contributes to cell adhesion interaction necessary for cell growth [43]. There is also a marked increase in the expression of HAS2 at the onset of tendon growth, with expression falling substantially by 28 days. As reported by Schwartz et al., in the original tendon, HA was located immediately adjacent to tenocytes, while HA was found throughout the neotendon ECM. By 7 days, the majority of the neotendon matrix stained positive for collagen, and at 28 days, nearly the entire neotendon area was occupied by collagen. While it is difficult to relate differences in abundance of a specific MMP with changes within the ECM, the stark upregulation in MMP13, and to a lesser extent in MMP2, MMP3, TIMP1, and TIMP2, suggests changes in proteolytic activity, which, along with increases in collagen expression and hydroxyproline content, suggest an active-matrix remodelling process [44]. Mitsui et al. reported the effect of HA on the expression of mRNAs for proinflammatory cytokines (IL-6, IL-1β, and TNF-α), and COX-2/PGE2 production in IL-1 stimulated subacromial synovial fibroblasts from patients with rotator cuff disease [45]. Aggrecan monomers form large aggregates consequent to their HA binding in the presence of Hyaluronan and proteoglycan link protein 1 (HAPLN1), resulting in considerable water absorption, making HA responsible for flexibility in animal tissues. While the quality in terms of size of HA in animal tissue decreases with age, its quantity increases [46]. HA is also a significant component of skin [47]: exposure of skin to ultraviolet rays causes reddening, and the derma stops HA production and increases its degradation [48]. HA is additionally helpful for the growth of epithelial tissue cells, eosinophils, and macrophages, and is also essential in healing and scar formation [49]. HA is degraded by a group of several enzymes named hyaluronidases. HA degradation products such as oligosaccharides and LMW-HA (<500 KDa) show proangiogenic properties [50]. HA is degraded under different conditions such as pH, temperature, mechanical, free radical, ultrasonic, and enzymatic stresses [51]. HA can also be degraded with non-enzymatic reactions such as acidic, alkaline hydrolysis, and oxidant decomposition [52]. ICAM-1 is considered a metabolic cell surface receptor of HA. It is involved in the clearance of HA from body fluid and plasma. Binding to this receptor triggers a coordinated cascade of events that induces endocytic vesicles fusion with primary lysosomes and catalyses its digestion to monosaccharides, transmembrane transport, phosphorylation, and catalyst deacetylation [53]. ICAM-1 can also act as a cell adhesion molecule, and the binding with HA might contribute to regulate ICAM-1-mediated inflammatory activation [54]. Edsfeldt et al. reported the role of PXL01, a synthetic peptide derived from lactoferrin, which exhibits an inhibitory effect on adhesion by reducing secretion of inflammatory cytokines, promoting fibrinolysis and reducing infections [55]. Tendon healing develops in three stages: haemorrhage–inflammation, proliferation–scar formation, and remodelling. Sodium hyaluronate (NaHA) has a strong effect on angiogenesis, as shown by VEGF and type IV collagen expression. Type IV collagen gradually accumulates in the subendothelial space, appearing in the early stage of angiogenesis and in the late stages of the healing process. Repetitive administration of NaHA demonstrated endothelial cell proliferation with strongly increased expression of VEGF [56].

## 3. Effects on Tendons

Overuse and traumatic events are capable of inducing remodelling of the tendon matrix through a mechanism that involves the degradation of collagen, the phagocytosis of the fragments obtained by the tenocytes, and a subsequent compensatory fibrillogenesis that restores the integrity of the tendon structure [57]. To our knowledge, the first experimental observation regarding the effects of HA on biomechanics and tendon repair dates back to 1980 and was performed on monkeys. St Onge et al. showed an improvement in the range of motion of the treated tendon, suggesting a possible role of HA in primary tendon repair [58]. Since then, the effect of HA application has been evaluated in numerous animal models, from rodents to primates, both in vitro and in vivo, and, more recently, in clinical trials with a limited number of participants [59]. To date, multiple mechanisms have been identified in the pathophysiological process of tendinopathies. HA had a beneficial effect on both the repair site and synovial sheath by decreasing the peripheral inflammatory response and promoting contact healing via involvement of epitenon and endotenon cells in the repair process [60]. Cell damage at epitenon level is evident—namely, necrosis, cells engaged in a phagocytic activity, massive destruction, and involution of large areas of the matrix [61]. HA 1% and 0.5% are more effective than triamcinolone in the prevention of adhesion and do not interfere with tendon healing [62]. Rapid change or an abrupt increase in the loading forces acting on the tendons can cause repeated microtraumas [63], capable of weakening the tendon structure in the long term. On the contrary, tendon disuse produces pathological variations of the collagen pattern similar to those of overuse [64]. Recently, Chisari et al. highlighted that the prolonged state of low-grade inflammation usually found in chronic tendinopathy can be considered as a risk factor for a “failed healing response” following an acute tendon insult [65]. In both cases, the tendon loses its functional and structural integrity with disruption of the fibres and reduction in resistance to mechanical stress. The first therapeutic approach consists of conservative treatment through the intake of anti-inflammatory drugs, the application of topical agents, injection of corticosteroids, and functional rehabilitation exercises of the affected joint. Recently, some clinical trials highlighted the beneficial effect of HA on tendon viscoelasticity [66]. The superiority of HA injections over other conservative treatments has been reported [67]. Several studies investigated the link between the inhibition of fibroblastic proliferation induced by HA, the stabilisation of type II collagen, and the reduction in type III collagen concentration in the tendon [68,69]. Currently, the ability of HA to stimulate new indirect synthesis of type I collagen is still being discussed, while its ability to increase cell viability and proliferation has been proven (Figure 2) [20]. Furthermore, HA seems to inhibit the expression of intermediate factors that play a key role in inflammatory pathways (NF-kB) in a dose-dependent manner [70]. This anti-inflammatory action is added to the ability of exogenous HA to reduce fragmentation of endogenous HA, stimulating new synthesis [71]. Regarding the therapeutic protocol, one or two high molecular weight HA injections are described in isolated tendinopathies for suitable short-term outcome [11,63].

## 4. Tissue Engineering and Tendon Healing

Regenerative medicine procedures are possible disease-modifying therapies for tendinopathies [72]. Several bioengineered systems have been developed using HA. Cells and growth factors alone cannot achieve optimal results in stimulating tenocytes differentiation without appropriate mechanical stimulation [73]. HA has been proposed as hydrogel or a 3D scaffold in combination with orthobiologics, such as tenoblasts, biomaterials, and growth factors to develop and support implantable tenocytes [74]. Scaffolds using HA are biodegradable, biocompatible, and bioabsorbable; HA plays a crucial role in cell signalling and cell growth, which can be attuned to the scaffold through its functional groups and functional domains [75,76]. HA hydrogel can be used as a carrier of drugs or biomaterials, combining it with epigallocatechin gallate (EGCG), a natural polyphenol with antioxidative and anti-inflammatory properties. This combination led to the mitigation of ischaemic and oxidative injury, and a suppression of the increased collagen III/I ratio in the tendinopathy group [72]. Recently, an esterified HA matrix (eHAM) has been used as a 3D scaffold to treat complicated lower extremity wounds with bone and tendon exposure with successful coverage and healing of exposed bone and tendons, because of the bridge role of eHAM providing a scaffold for fibroblast and endothelial cells and stimulating angiogenesis [77]. The use of a 3D scaffold has a double function: first, HA scaffold can carry biomaterials and growth factors into the tendinopathic tendon; then, it provides the mechanical stimulation necessary to tenocytic differentiation [78,79]. Ciardulli et al. investigated the effects of the HY-FIB 3D HA scaffold on human bone marrow mesenchymal stem cells (hBM-MSCs) in static and dynamic scenarios, showing a significant increase in tenogenic differentiation and anti-inflammatory cytokines production in dynamic conditions [80]. HA has been used also as a resorbable, suturable, and biocompatible mesh in the treatment of neglected Achilles tendon rupture. In 2014, Esenyel et al. applied Hyalonect as a reinforcement of a turndown gastrocnemius–soleus fascial flap, reporting an excellent or good return of function after surgery [81].

## 5. Tendinopathies and HA Clinical Applications

Tendinopathies are increasingly common, negatively impacting patients’ quality of life [82]. Several intrinsic and extrinsic risk factors can contribute to the pathogenesis of tendinopathies, including mechanical overload, poor vascularity, age, gender, and genetic, endocrine, and metabolic factors [83,84,85]. Tendinopathic tendons show diffuse structural changes: increased tenocyte apoptosis, disruption of collagen fibres with decreased collagen type I production, disorderly increased type III collagen production, and ineffective neoangiogenesis [83]. Patients report pain at the tendon site which increases with exercise and everyday life activities and limits sports performances [86]. Other clinical signs are local tendon tenderness, limited range of motion, and swelling [87]. The management of tendinopathy is debated, and there is still no consensus about a gold standard treatment. However, HA could have several disease-modifying effects, leading to increased tenocytes regeneration, restoration of collagen type I/type III ratio, reduced apoptosis, and angiogenetic changes [18,80,81].

## 6. Rotator Cuff Tendinopathy and HA

Rotator cuff tendinopathy is the most common cause of shoulder pain, with an increasing frequency that varies from 5% to 40%, especially affecting the supraspinatus tendon [88]. Recently, several authors compared HA injections with different management options. Meloni et al. evaluated the effects of ultrasound-guided HA injections in supraspinatus tendinopathy HA induced symptoms and disability improvement, compared with placebo, up to 9 months follow-up [89]. Merolla et al. compared ultrasound-guided subacromial injections of HA with physical therapy. Both treatments lead to pain relief and clinical improvement in the short term, but the HA group maintained a significant improvement at 12 weeks of follow-up. Frizziero et al. in a prospective study investigated the effects of low molecular weight HA injections in the subacromial space compared with extracorporeal shockwaves therapy in patients with non-calcific supraspinatus tendinopathy. This study confirmed Merolla et al.’s findings, showing that one low molecular weight HA injection a week for three weeks produced clinical relief at the end of the treatment and for up to 3 months. No significant difference was found between the HA injections and extracorporeal shockwave therapy in terms of safety and efficacy [67]. Comparing physical therapy alone or combined with HA injections in the management of supraspinatus tendinopathy, Flores et al. found that patients treated with physical therapy and HA returned significantly earlier to work and needed fewer rehabilitation sessions [90].

## 7. Patellar Tendinopathy and HA

Patellar tendinopathy is common in sports in which athletes jump, as in volleyball, basketball, or triple jump [91]. Kumai et al. evaluated the effects of a single high molecular weight HA injection in patients with patellar tendinopathy, finding improvement in pain and visual analogue scale (VAS) values at short-term follow-up (one week) [13]. Muneta et al. treated 50 young athletes with two high molecular weight HA injections without ultrasound (US) guidance. This was effective, safe, and repeatable [92]. Fogli et al. showed that a cycle of one HA injection a week for three weeks was effective and safe in patellar tendinopathy, with a significant decrease in pain, mean VAS values, and improvement of US appearance [93]. Kaux et al. compared US-guided injections of platelet-rich plasma (PRP) versus two HA injections, reporting that both treatments could be effective. The PRP group showed significant improvement in quadriceps strength, while HA had a greater impact on the improvement of symptoms [94]. Recently, Frizziero et al. showed good results after three medium molecular weight US-guided HA injections. This study reported pain relief and improvement of the Victorian Institute of Sport Assessment for the patellar tendon (VISA-P) values at 90 days follow-up, with a decrease in vascularisation and tendon thickness at US and Power Doppler analysis [10].

## 8. Achilles Tendinopathy and HA

Achilles tendinopathy affects mainly athletes, especially runners, but also non-athletes [95]. The therapeutic use of HA in Achilles tendinopathy has been recently described (Figure 3A,B). Lynen et al. compared the effects of two HA peritendinous injections with shockwave therapy in patients with mid-portion Achilles tendinopathy. At 6 months, patients treated with HA injections reported better outcomes, with greater symptom improvement and restored function [14]. Similarly, Fogli et al. and Frizziero et al. showed that three US-guided medium molecular weight HA injections induce a clinically relevant increase in VISA-A values, pain relief, and US parameters improvement [10,93]. Recently, Gervasi et al. investigated the clinical, viscoelastometric, and biochemical effects of three US-guided medium molecular weight HA injections in runners with unilateral Achilles tendinopathy. Indeed, patients reported improvement in clinical assessment, decreased pain and stiffness of the tendon, and reduction in the viscoelastometric and functional asymmetry between the affected tendon and the healthy limb [96,97].

## 9. Epicondylitis and HA

Lateral epicondylitis is a common cause of chronic elbow pain and affects 1% to 3% of the general population per year [98]. The tendon structures most affected are the insertions of the extensor of the forearm, located on the lateral side of the elbow [98]. Petrella et al. compared outcomes in patients who received HA injections versus a control group, who received an injection of 1.2 mL of a saline placebo, finding significantly greater improvement in VAS pain at rest and after grip testing up to 1-year follow-up [99]. Khan et al. reported the efficacy of a single HA injection in the management of moderate epicondylitis (VAS pain score < 7), while it was not effective in severe lateral epicondylitis [100]. In a prospective randomised trial, Tosun et al. compared a combined HA-chondroitin sulphate injection versus a corticosteroid injection, showing an equal reduction in pain and improvement of function in the short term, while HA resulted in better outcomes at long-term follow-up [101].

## 10. Conclusions

Anti-inflammatory, wound healing, antiangiogenic, and immunosuppressive effects of HA have been reported in in vitro and in vivo studies. These findings encourage the clinical application of HA in tendinopathies such as rotator cuff, epicondylitis, Achilles, and patellar tendinopathy. However, there are still questions to be answered and issues to be addressed. First of all, many aspects of HA metabolism, receptor clustering, and affinity still need to be explored to understand the different biological actions that hyaluronan has in the inflammatory process also in relation to molecular weight. Understanding these mechanisms could provide opportunities to extend and improve hyaluronan pharmaceutical, biomedical, cosmetics, and food supplements applications, obtaining more targeted effects.

## Figures and Tables

**Figure 1 cells-10-03081-f001:**
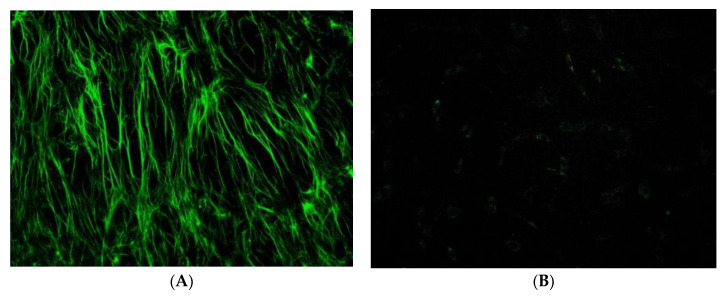
(**A**) Type I collagen in tenocytes, harvested from degenerated human supraspinatus tendon, stimulated for 14 days with 1000 μg/mL (>500 KDa) of HA; (**B**) untreated cells.

**Figure 2 cells-10-03081-f002:**
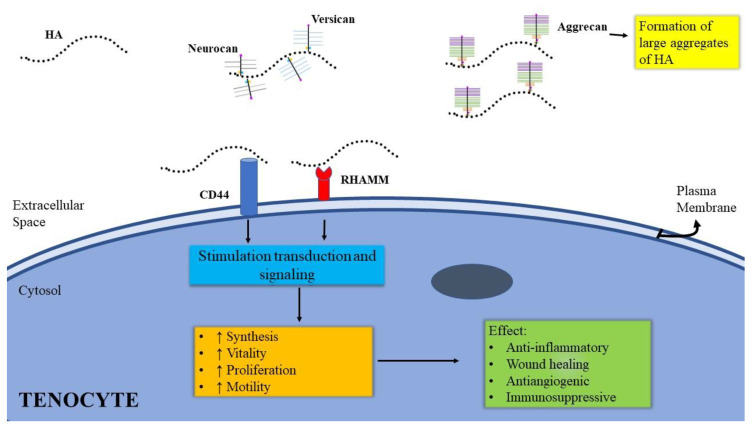
Effects of HA on tenocytes.

**Figure 3 cells-10-03081-f003:**
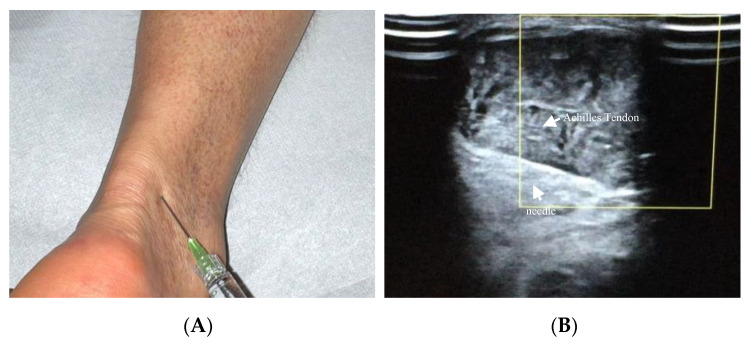
(**A**) Injection of HA in a dorsolateral approach of Achilles tendon; (**B**) US visualisation (5–12 MHz linear probe and PRF set at 0.5 kHz) of the needle (22-gauge) introduced at a 30-degree angle in the mesotendon, with the probe in a transverse plane.

**Table 1 cells-10-03081-t001:** ECM components of tendons.

ECM Components	%
Collagen	86% (type I: 98%)
Proteoglycan	1–5%
Elastin	2%
Decorin	<1%
Aggrecan	<1%
Other proteins	<1%

**Table 2 cells-10-03081-t002:** HA synthesis enzymes; HMW-HA, high molecular weight HA; MMW-HA, medium molecular weight HA; LMW-HA, low molecular weight HA; HAS, HA synthases.

HA SYNTHASES	
HMW-HA and LMW-HA	HAS1
	HAS2
LMW-HA	HAS3

**Table 3 cells-10-03081-t003:** HA interactions. RHAMM, receptor for hyaluronan-mediated motility; LYVE-1, lymphatic vessel endothelial hyaluronan receptor 1; TLR-4, Toll-like receptors 4; TSG6, TNF-stimulated gene 6; GHAP, glial hyaluronate-binding protein; LMW-HA, low molecular weight HA.

HA and ECM Receptors Interactions	
	Function
CD44 Receptor for hyaluronan-mediated motility (RHAMM)	Binding HA: anti-inflammatory, wound healing, antiangiogenic, immunosuppressive
Lymphatic vessel endothelial hyaluronan receptor 1 (LYVE-1)	Lymphatic transport of leukocytes
Toll-like receptor 4 (TLR-4)	Binding LMW-HA: pro-inflammatory
TNF-stimulated gene 6 (TSG-6)	Tissue protective and anti-inflammatory
Glial hyaluronate-binding protein (GHAP)	Reduces the spread of inflammatory cells in nerve tissue
Neurocan Versican Aggrecan	Development, cell migration, maturation and differentiation, cell survival, and tissue homeostasis

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
