# Peer review of "The Impact of Hyaluronic Acid on Tendon Physiology and Its Clinical Application in Tendinopathies"

_cells, 2021, doi:10.3390/cells10113081_

Round 1

Reviewer 1 Report

Thank you very much for giving me the opportunity to review the manuscript titled: “The impact of hyaluronic acid on tendon physiology and its clinical application in tendinopathies” for your renowned journal.

The authors’ aim was to perform a narrative review in order to investigate the current knowledge on the biological properties and clinical efficacy of hyaluronic acid in tendinopathies. The authos state that the physical-chemical, structural, hydrodynamic biological properties of hyaluronic acid are still poorly investigated while medical history and clinical applications are still debated. Several pre-clinical and clinical reports are said to show a good efficacy and safety profile of hyaluronic acid, despite the lack of consistency in literature regarding the classification according to molecular weight. In in-vitro and pre-clinical studies, hyaluronic acid is described to have shown physical-chemical properties, such as biocompatibility, mucoadhesivity, hygroscopicity and viscoelasticity, useful to contribute to tendon healing. Also, in clinical studies, hyaluronic acid is reported to be used with encouraging results in different tendinopathies of various types and anatomical sites. The narrative review on hand intends to update current knowledge on the biological properties and clinical efficacy of hyaluronic acid in tendinopathies.

This paper proved relevant and interesting to me as a reviewer. There are only some minor comments, which have to be revised prior to potential publication on my viewpoint. Please find my detailed comments hereby listed:

Title:

Appropriate.

Abstract:

Please consistently adhere to a spelling of “hyaluronic acid” in the abstract, either all capitalized or lower-case spelling.

Room for improvement regarding concision; maybe structuring the abstract with subheadings (“Background”, “Results”, “Conclusion”). Adding the conclusion would be of interest here.

Line 22: It is unclear as to what is meant by “great confusion” regarding molecular weight, please specify or adapt wording.

Lines 26-27: The phrasing “different tendinopathies in terms of type and anatomical site” could maybe be more concise.

Introduction:

Line 39: Please specify the abbreviation (ECM) when the word is first used (Line 35).

Line 42: “Predominant” is spelled without a hyphen.

I suggest restructuring the introduction paragraph by describing the relevance of HA injections last and adding the purpose of the paper in the course of this.

HA Synthesis, Properties, and Degradation:

Lines 78: Please specify the statement: “HA acts on the ECM by specific and non-specific interactions.”

Line 96: Please specify the abbreviation (TLR) when the word is first used.

Line 104: Please specify the abbreviation (GHAP) before introducing it.

In general, I suggest adding paragraphs to a lengthy chapter like this for better overview and structure.

Effects On Tendons:

Line 171: Please specify which conservative treatments are inferior to HA injections.

Paragraphs could provide a better overview of this chapter. Starting with line 196, clinical applications are discussed, which might be more appropriate for the following chapter.

Tendinopathies And HA Clinical Applications:

Appropriate chapter.

Rotator Cuff Tendinopathy and HA:

Line 226: Please specify the reported incidence number. As incidence is defined as a measure of frequency expressed over a specified time period, the calculation of the percentage given is unclear.

Patellar Tendinopathy and HA:

Please specify all abbreviations before using continuously them in the text. Otherwise, appropriate chapter.

Achilles Tendinopathy and HA:

Line 263: “… has been recently described” – please specify reference and give examples. Instead of referencing an article in the subtext of Figure 3, I would advice keeping the subtext short and poignant and transferring the longer text to the intended paragraph.

Epicondylitis and HA:

Appropriate and concise.

Conclusions:

Generally appropriate and well-written.

Line 292: “However, there are still questions to be answered and issues 292 to be addressed.” – Please specify or use references.

Line 295: Molecular weight is addressed here, yet hardly discussed in the main body of the article. It would be interesting to read more about its relevance, for example in the chapter regarding its biological properties.

A section about general limitations of the research and data provided would provide more insight in this matter and acknowledge a potential risk of bias.

Abbreviations:

Please make sure to specify each term before abbreviating it. In the light of the many abbreviations used in this paper, it could perhaps be useful to create a separate table of abbreviations.

References:

Relevant.

Tables and figures:

Overall providing relevant information. Table 2 is not referenced in the text and its relevance remains unclear. The fonts in Table 1 and in Figure 2 could possibly be adapted to match the style of the other font used.

General note:

Please proof-read the text for general English, as it is not always grammatical. For example, in Line 34 it is unclear what is meant by “widely express”; in line 104, there is an article missing (“is an ECM protein” vs. “is ECM protein”).

Please add a systemtic literature search according to the PRISMA criteria or explain, why you did not choose doing that.

Thank you again for allowing me the opportunity to review this work for your journal.

With kind regards,

Patrick Sadoghi, MD, PhD, MBA

Author Response

Ref.: Cells-1414921

Thank you for giving us the opportunity to revise the manuscript "The impact of hyaluronic acid on tendon physiology and its clinical application in tendinopathies" to the Cells. The comments of the Reviewers have been carefully considered, and implemented as follows:

 Reviewer #1

Title: Appropriate.

Abstract: Please consistently adhere to a spelling of “hyaluronic acid” in the abstract, either all capitalized or lower-case spelling.

Room for improvement regarding concision; maybe structuring the abstract with subheadings (“Background”, “Results”, “Conclusion”). Adding the conclusion would be of interest here.

Line 22: It is unclear as to what is meant by “great confusion” regarding molecular weight, please specify or adapt wording.

Lines 26-27: The phrasing “different tendinopathies in terms of type and anatomical site” could maybe be more concise.

 ANSWER  thank you for raising these points. All the reviewer’s suggestions have been applied and modified through the manuscript.

Introduction: Line 39: Please specify the abbreviation (ECM) when the word is first used (Line 35).

Line 42: “Predominant” is spelled without a hyphen.

I suggest restructuring the introduction paragraph by describing the relevance of HA injections last and adding the purpose of the paper in the course of this.

ANSWER  thank you for raising these points. All the reviewer’s suggestions have been applied and modified through the manuscript.

HA Synthesis, Properties, and Degradation:

Lines 78: Please specify the statement: “HA acts on the ECM by specific and non-specific interactions.”

Line 96: Please specify the abbreviation (TLR) when the word is first used.

Line 104: Please specify the abbreviation (GHAP) before introducing it.

In general, I suggest adding paragraphs to a lengthy chapter like this for better overview and structure.

 ANSWER  thank you for raising these points. All the reviewer’s suggestions have been applied and modified through the manuscript.

Effects On Tendons:

Line 171: Please specify which conservative treatments are inferior to HA injections.

ANSWER thank you for raising this point. The conservative treatments compared to HA injections have already been cited in the sentence before line 171.

Tendinopathies And HA Clinical Applications: Appropriate chapter.

ANSWER  thank you for your comment.

Rotator Cuff Tendinopathy and HA:

Line 226: Please specify the reported incidence number. As incidence is defined as a measure of frequency expressed over a specified time period, the calculation of the percentage given is unclear.

ANSWER  thank you for raising this point. Frequency has been added and used instead of incidence to give a more accurate information.

Patellar Tendinopathy and HA:

Please specify all abbreviations before using continuously them in the text. Otherwise, appropriate chapter.

Achilles Tendinopathy and HA:

Line 263: “… has been recently described” – please specify reference and give examples. Instead of referencing an article in the subtext of Figure 3, I would advice keeping the subtext short and poignant and transferring the longer text to the intended paragraph.

ANSWER  thank you for raising these points. The Achilles tendinopathy paragraph has been reorganized as recommended by the reviewer.

Epicondylitis and HA: Appropriate and concise.

Conclusions: Generally appropriate and well-written.

Line 292: “However, there are still questions to be answered and issues 292 to be addressed.” – Please specify or use references.

ANSWER  thank you for raising this point. It has been specified.

Abbreviations: Please make sure to specify each term before abbreviating it. In the light of the many abbreviations used in this paper, it could perhaps be useful to create a separate table of abbreviations.

ANSWER thank you for raising this pont. All the abbreviation have been specified in the body of the text.

References: Relevant.

Tables and figures: Overall providing relevant information. Table 2 is not referenced in the text and its relevance remains unclear. The fonts in Table 1 and in Figure 2 could possibly be adapted to match the style of the other font used.

ANSWER  thank you for raising these points. All the reviewer’s suggestions have been applied and Figures and Tables have been modified as requested.

General note:

Please proof-read the text for general English, as it is not always grammatical. For example, in Line 34 it is unclear what is meant by “widely express”; in line 104, there is an article missing (“is an ECM protein” vs. “is ECM protein”).

ANSWER thank you for raising this point. All the reviewer’s suggestions have been applied and modified through the manuscript.

Please add a systemtic literature search according to the PRISMA criteria or explain, why you did not choose doing that.

ANSWER  thank you for raising this point. PRISMA criteria have not been applied because of the authors were invited to a narrative review, instead of conducting a systematic review of the literature.

Reviewer 2 Report

The review by Oliva et al. describes the impact of hyaluronic acid on tendon physiology and its clinical application in tendinopathies. The concise review is written comprehensibely and gives a good overview on the current knowledge on structure and function of hyaluronic acic and its possible positive action on tendon physiology in the context of diseased tendon healing. Still, some parts need to be revised before publication.

Line 65-66: please add the respective molecular weight for HMW-HA and LMW-HA

Line 108-115: Please revise this paragraph as it appears somewhat out of context, so that for the reader it is difficult to understand what is meant with „original tendon“, „onset of tendon growth“, and „neotendon“.

Line 184: This sentence lacks a beginning.

Please revise all figure legends and add references or source links.

Figure 1: The image showing collagen 1 staining in tenocytes after stimulation with HA is not very informative per se. I suggest to add a control image to better demonstrate alterations induced after HA- incubation. Authors should describe the difference seen.

Figure 2: This figures illustrates the proteins involved in HA signalling. Addition of specific signalling pathways, if known, would be informative.

Please add a reference or source link to every figure.

Figure 3:  This figure may be omitted since it does not add much information. I recommend to integrate the text of the legend into to the main text.

Figure 4: Please add arrows to indicate what can be seen on the US image as it gets not clear for a reader not being expert in ultrasound imaging (bone, tendon, site of injection…) The image is missing technical information such as frequency, scale, etc. Please add to the figure legend. The text of the legend referring to studies on Achilles tendinopathy should be integrated into the main text as recommended above.

Line 297: The authors state in their conclusions that a better understanding of these mechanisms could extend and improve cosmetics and food applications. It would be interesting to know which studies the authors refer to?

„Understanding these mechanisms could provide opportunities to extend and improve hyaluronan pharmaceutical, biomedical, cosmetics and food supplements applications, obtaining more targeted effects.“

Author Response

Ref.: Cells-1414921

Thank you for giving us the opportunity to revise the manuscript "The impact of hyaluronic acid on tendon physiology and its clinical application in tendinopathies" to the Cells. The comments of the Reviewers have been carefully considered, and implemented as follows:

Reviewer #2

Line 65-66: please add the respective molecular weight for HMW-HA and LMW-HA

Line 108-115: Please revise this paragraph as it appears somewhat out of context, so that for the reader it is difficult to understand what is meant with „original tendon“, „onset of tendon growth“, and „neotendon“.

Line 184: This sentence lacks a beginning.

Please revise all figure legends and add references or source links.

ANSWER  thank you for raising this point. All the reviewer’s suggestions have been applied and modified through the manuscript.

Figure 1: The image showing collagen 1 staining in tenocytes after stimulation with HA is not very informative per se. I suggest to add a control image to better demonstrate alterations induced after HA- incubation. Authors should describe the difference seen.

ANSWER  thank you for raising this point. Unfortunately, this image comes from a laboratory experience and there is no control image available at this moment. 

Figure 2: This figures illustrates the proteins involved in HA signalling. Addition of specific signalling pathways, if known, would be informative.

ANSWER  thank you for raising this point. All the known pathways have been reported in the image.

Figure 3:  This figure may be omitted since it does not add much information. I recommend to integrate the text of the legend into to the main text.

Figure 4: Please add arrows to indicate what can be seen on the US image as it gets not clear for a reader not being expert in ultrasound imaging (bone, tendon, site of injection…) The image is missing technical information such as frequency, scale, etc. Please add to the figure legend. The text of the legend referring to studies on Achilles tendinopathy should be integrated into the main text as recommended above.

ANSWER  thank you for raising this point. All the reviewer’s suggestions have been applied and modified through the manuscript. However arrows have not been added because it would have altered the figure quality since it is a small real-time US image.

Line 297: The authors state in their conclusions that a better understanding of these mechanisms could extend and improve cosmetics and food applications. It would be interesting to know which studies the authors refer to?

„Understanding these mechanisms could provide opportunities to extend and improve hyaluronan pharmaceutical, biomedical, cosmetics and food supplements applications, obtaining more targeted effects.“  

ANSWER  thank you for raising this point. The authors idea is that future studies are need to improve HA use.

Reviewer 3 Report

The article gives an interesting overview over the pecularities of HA and explains the impact of HA on tendon with a focus on tendinopathy. It connects cell biological and biochemical knowledge with some clinical experiences of HA treatment in tendon. The different effects of cross-linked and less stabilized HA could be added in more detail since there exist a plethora of different compounds suggested for therapy. The authors should try to link each issue closely to tendon (e.g. is RHAMM expressed by tenocytes?). Some unclearnesses should be adressed (issues arer listed below). Please check that all abbreviations used are indeed explained with their first use.

line 34: write „expressed“

line 37: twotimes „structures“ substitute by „component“ and the second one by „tissue“.

Line 39: „migration“ could also be mentioned

Line 44: „tendon structures“ better to write „tendon tissue“

„tenoblast-tenocyte“, better to write „tenoblast or tenocyte“, the difference should be explained

Line 45: „consist of“, better to write „include“ instead

„all cells of the ECM“ since the ECM is a product of cells it makes no sense

Line 50: „ineffective neovascularization“, please explain it  (see also page 218)

Legend of figure 1_ from which tendon derive the tenocytes shown? How much HA was administered

Table 1: please revise, decorin and aggrecan belong to the proteoglycan fraction?

Line 69: do there exist MMPs which cleave HA?

Line 80:  I would say that aggrecan is not a typical receptor since it represents an ECM component

Line 83: not only chondroitin sulfate but also keratan sulfate

Line 88: „Aggrecan…“ please mention that it plays also a role in tendon

Line 93: cite a reference and explain whether RHAMM is also expressed in tendon

Line 108: „original“ what means it here?

Line 109: „HA….HA“ this sentence is somehow unclear

Line 141: „proliferation-scar“ I would not include the scare as a healing phase in tendon. Scar is an unwanted outcome, but not a natural phase in healing

Line 145: „NaHA“ please explain this abbreviation

Table 2 and 3: explain the abbreviations

Line 156: „numerous animal models“ of tendon healing?

First line on page 6: start the sentence with capital letters

Line 216: „collagen I“, write „collagen type I“.

Page 7: explain abbreviation at their first use e.g. US, PRP, VAS, VISA-P

Figure 2: „stimulation transduction and signalling“ which pathways are induced by HA binding?

The NFkB pathway could be mentioned

Legend of Figure 3: write „achilles tendon“ with capital letter, a point behind it is lacking.

Line 277: „epicondylitis“ please state which tendon(s) are generally affected.

Figures 3 and 4 could be combined as Fig. 3A+B.

Author Response

Ref.: Cells-1414921

Thank you for giving us the opportunity to revise the manuscript "The impact of hyaluronic acid on tendon physiology and its clinical application in tendinopathies" to the Cells. The comments of the Reviewers have been carefully considered, and implemented as follows:

Reviewer n#3

Line 34: write „expressed“

line 37: twotimes „structures“ substitute by „component“ and the second one by „tissue“.

ANSWER  thank you for raising this point. All the reviewer’s suggestions have been applied and modified through the manuscript.

Line 39: „migration“ could also be mentioned

ANSWER  thank you for raising this point. All the reviewer’s suggestions have been applied and modified through the manuscript. However, we do not consider it necessary, for the purposes of the article, to deepen this topic.

Line 44: „tendon structures“ better to write „tendon tissue“

„tenoblast-tenocyte“, better to write „tenoblast or tenocyte“, the difference should be explained

Line 45: „consist of“, better to write „include“ instead „all cells of the ECM“ since the ECM is a product of cells it makes no sense

ANSWER  thank you for raising this point. All the reviewer’s suggestions have been applied and modified through the manuscript.

Line 50: „ineffective neovascularization“, please explain it.

ANSWER  thank you for raising this point. It has already been reported in the current literature and represent a structural change show in tendinopathy.

Legend of figure 1_ from which tendon derive the tenocytes shown? How much HA was administered.

ANSWER thank you for raising these points. The reviewer’s suggestions have been applied and the figure 1 legend has been modified.

Table 1: please revise, decorin and aggrecan belong to the proteoglycan fraction?

ANSWER  thank you for raising this point. They are part of the proteoglycan fraction. In the table they have been inserted separately to underline their percentage, relatively high compared to all proteoglycans, compared to the other components of the ECM

Line 69: do there exist MMPs which cleave HA?

ANSWER  thank you for raising this point. There are no MMPs that degrade HA. It has been observed in some experimental studies that HA is able to block the action of fibronectin fragments that activate MMPs

Line 80:  I would say that aggrecan is not a typical receptor since it represents an ECM component

In the period, both the receptors and the molecules that interact with HA are listed.

Line 83: not only chondroitin sulfate but also keratan sulfate

Line 88: „Aggrecan…“ please mention that it plays also a role in tendon

Line 93: cite a reference and explain whether RHAMM is also expressed in tendon

ANSWER  thank you for raising this point. All the reviewer’s suggestions have been applied and modified through the manuscript.

Line 108: „original“ what means it here?

ANSWER  thank you for raising this point. The original term refers to the normal tendon structure before a possible injury and the beginning of the healing phase

Line 109: „HA….HA“ this sentence is somehow unclear

ANSWER  thank you for raising this point. The line has been modified in order to make it clearer.

Line 141: „proliferation-scar“

ANSWER  thank you for raising this point. The authors would not include the scare as a healing phase in tendon. Scar is an unwanted outcome, but not a natural phase in healing. We consider it the appropriate term because it is a possible evolution of the healing process.

Line 145: „NaHA“ please explain this abbreviation. Sodium hyaluronate (NaHA)

Table 2 and 3: explain the abbreviations

Line 156: „numerous animal models“ of tendon healing?

First line on page 6: start the sentence with capital letters

Line 216: „collagen I“, write „collagen type I“.

Page 7: explain abbreviation at their first use e.g. US, PRP, VAS, VISA-P

ANSWER  thank you for raising this point. All the reviewer’s suggestions have been applied and modified through the manuscript.

Figure 2: „stimulation transduction and signalling“ which pathways are induced by HA binding?

ANSWER  thank you for raising this point. There are numerous pathways like NFkB, Src, p185HER2 and other but we do not consider it necessary to elaborate on this aspect in our article.

Legend of Figure 3: write „achilles tendon“ with capital letter, a point behind it is lacking.

Line 277: „epicondylitis“ please state which tendon(s) are generally affected.

Figures 3 and 4 could be combined as Fig. 3A+B.

ANSWER thank you for raising this point. All the reviewer’s suggestions have been applied and modified through the manuscript.

Round 2

Reviewer 3 Report

The manuscript has been sufficiently revised, my previous comments have been addressed.